# *My Journey*: Development and Practice-Based Evidence of a Culturally Attuned Teen Pregnancy Prevention Program for Native Youth

**DOI:** 10.3390/ijerph16030470

**Published:** 2019-02-06

**Authors:** DenYelle Baete Kenyon, Tracey R. McMahon, Anna Simonson, Char Green-Maximo, Ashley Schwab, Melissa Huff, Renee E. Sieving

**Affiliations:** 1Behavioral Sciences, Sanford Research, Sioux Falls, SD 57104, USA; tracey.mcmahon@sanfordhealth.org (T.R.M.); anna.simonson@sanfordhealth.org (A.S.); char.green@sanfordhealth.org (C.G.-M.); 2Department of Pediatrics, Sanford School of Medicine, University of South Dakota, Sioux Falls, SD 57105, USA; 3Marketing, Sanford Health, Sioux Falls, SD 57104, USA; ashley.schwab@sanfordhealth.org; 4Sisseton-Wahpeton Oyate Tribal Health Administration, Agency Village, SD 57262, USA; melissahuff4@hotmail.com; 5School of Nursing & Department of Pediatrics, University of Minnesota, Minneapolis, MN 55455, USA; sievi001@umn.edu

**Keywords:** American Indian/Alaska Native, pregnancy prevention, adolescents, positive youth development, curriculum development

## Abstract

A clear need exists for teen pregnancy prevention programs that are responsive to the specific needs and cultural contexts of Native American communities. Recent data indicates that the birth rate for Native teens is nearly two and a half times the rate for White teens (32.9 versus 13.2). To address this disparity, we conducted formative research with Northern Plains Native American community members, resulting in *My Journey*, a culturally attuned curriculum for 6–8^th^ graders. *My Journey* is grounded in traditional values and teachings to promote self-efficacy in sexual health decision-making and engagement in prosocial behaviors. We conducted a pilot study with 6–8^th^ grade students (*n* = 45), aged 11–14 years (22 females, 23 males). Pilot study findings confirm program feasibility and acceptability. The process evaluation revealed that teachers liked the curriculum, particularly its adaptability of cultural components and ease of student engagement. The outcome evaluation demonstrated that *My Journey* provided an avenue for NA youth to increase their sex refusal self-efficacy. Application of the culture cube framework revealed *My Journey* has made a meaningful practice-based evidence contribution as a community-defined, culturally integrated curriculum that is effective. Future directions include broader implementation of *My Journey*, including adaption for additional populations.

## 1. Introduction

The teen birth rate (births per 1,000 15-19 year old females) has steadily declined over the past decade, with a decrease of 55% from 2007 (41.5) to 2017 (18.8), a historic low for the United States (U.S.) [1]. The decline is largely attributed to the fact that, more teens are 1) using contraceptives, and are 2) using them more effectively [2], which is likely, at least in part, due to teen pregnancy prevention efforts [2,3]. Although decreases in teen birth rates have occurred for all racial and ethnic groups, rates in the U.S. remain particularly high among industrialized countries, and racial and ethnic disparities persist [4]. In 2017, the teen birth rate for Native Americans (NA) remained two and a half times as high as the rate for Whites, and nearly twice as high as the national average. In fact, in 2015, there was a sharp increase in the birth rate for Native teens (see Figure 1) [1].

### 1.1. Pregnancy Risk Behaviors among Native American Youth

Previous research suggests several reasons why teen pregnancy rates among Native American adolescents are higher. Youth Risk Behavior Survey (YRBS) results reveal that, compared to their peers, a greater proportion of Native youth reported a cluster of related risk behaviors, including early sexual debut, sex with multiple lifetime partners, dating violence, forced sex, early alcohol use, and early drug initiation and lifetime drug use [6,7]. Despite a lack of information regarding contraceptive use among Native teens, related data, such as teen pregnancy rates, strongly suggest that Native teens are also less likely to use contraception when compared to other racial and ethnic groups, indicating a need for teen pregnancy prevention interventions to address sexual risk behaviors and access to contraceptives for Native youth specifically [8,9,10]. 

Our previous research with Native American communities about birth control and sex helps elucidate why Native teens may be less likely to use contraception. For instance, Northern Plains Native youth stated that many Native teens believe birth control is a woman’s responsibility, not a man’s, because women, it is supposed, are “more aware” and “more responsible” [11]. Additionally, many Native youth have misbeliefs about the side effects of certain birth control methods, making contraceptive use among Native teens a complicated issue. For girls who might want to use contraception, there is often little or no opportunity to discuss it with their male sex partners. Many NA boys learn to view sex competetively from a young age, as a “notch in their belt” or a “badge of honor”. In such an environment, girls frequently have sex out of pressure, fear, or force [11].

### 1.2. Practice-Based Evidence

The unique cultural context within which such beliefs about sex and contraception develop emphasizes the need for innovative, culturally-specific sexual health curriculums for Native youth. Although a number of efforts have built on the assets and strengths of NA cultures to improve the reproductive health of Native youth (e.g., Native STAND, Circle of Life, Project Red Talon), there remains a dearth of culturally attuned programs for this at-risk population [12]. Evidence is undoubtedly growing for recent work; but to date, there are no programs specific to Native youth on the U.S. Department of Health and Human Services Evidence-based Teen Pregnancy Prevention programs list [13,14]. A 2016 article in the American Journal of Public Health noted the absence of teen pregnancy prevention curriculums for Native American communities, suggesting that the underlying issue was related to the “challenges of implementing rigorous evaluation designs within tribal communities” [15]. An underlying issue may in fact be that researchers are trying to force culturally and ethnically diverse communities into the homogenous, oftentimes extraneous frameworks of Western evidence-based science. As Hawkins and Walker (2005) have argued, “empirically-based science leaves no room for the cultural context that is crucial to the success of a treatment approach within tribal communities” [16]. Increasingly, researchers who work with diverse and underrepresented populations have acknowledged the difficulties that arise when traditional Western practices are grafted onto communities for whom the evidence doesn’t apply, and overly specific and/or limited definitions of evidence are problematic [17]. In response, some have called for the inclusion of more innovative approaches to intervention program development with non-majority populations, including the development of culturally specific, community-driven teen pregnancy prevention programs [18].

Our work utilizes a practice-based evidence (PBE) framework, “a range of treatment approaches and supports that are derived from, and supportive of, the positive cultural attributes of the local society and traditions,”Isaacs et al. [19]. Much of the current research on evidence-based methodologies with non-majority populations argues that a mixed approach is ideal [17,20,21,22,23,24]—combining scientific rigor with the more nuanced and culturally-responsive methods of PBE. However, conjoining PBE with traditional Western methods has proven difficult, in large part, because Western empiricism continues to be “uncritically accepted as the dominant paradigm over indigenous epistemologies,” as Abe, Grills, Ghavami, Xiong, Davis and Johnson [17] recently highlighted. 

For researchers who work with Native American communities and other non-majority populations, the evidence is ideally derived from the community itself, beginning with the formative research, when researchers ask community members and stakeholders to identify community needs and interventions that would likely yield the most positive results. Indeed, community-based participatory research (CBPR) goes hand-in-hand with PBE, as many researchers have acknowledged [21,25], and is critical to the effective development of intervention programs within culturally and ethnically diverse groups. CBPR expands the definition of evidence beyond empirically-based controlled trials, to evidence as a community sees fit [26]. “Values, beliefs, assumptions, and practices are culturally, historically, and contextually determined and can be clearly articulated, understood, and evaluated,” as Abe et al argues [17]. Over time, community members have determined what works and what doesn’t within their population groups, and, whether or not this evidence has been empirically measured, certain practices are deemed effective because of the positive results they yield [26]. PBE is critical, as Lucero [27] has argued, for the “decolonization” of methodologies that privilege Western science over Indigenous ways of knowing. 

#### Culture Cube

The inclusion of culture is also critical in the development of PBE intervention programs, and requires the full participation of community members and stakeholders in order for a program to be both accepted and effective. As Walsh-Buhi [24] titled her article on the PBE for substance abuse prevention programs for Native American communities, “Please don’t just hang a feather on a program or put a medicine wheel on your logo and think ‘oh well, this will work.’” To fully incorporate culture into PBE programs, three critical criteria must be addressed: “(1) information about desired goals and outcomes comes directly from the people receiving services; (2) cultural factors can be explicitly included in interventions; and (3) effectiveness can be measured according to these outcomes”[20]. 

The culture cube is a conceptual tool developed by Abe, Grills, Ghavami, Xiong, Davis and Johnson [17] “to identify and articulate the cultural underpinnings of prevention and early intervention projects in five priority populations,” including Native Americans. The culture cube can be used either with existing PBEs, or as a way to develop culturally-conscious PBEs where none currently exist [17]. The culture cube consists of six elements, three visible (project, place, and persons), and three invisible (culture, causes, and changes), the latter three of which represent the cultural underpinnings of a particular PBE. There are three recognized uses of the cube: “(a) identifying and articulating how culture shaped these projects, (b) how cultural issues could guide the development of evaluation questions, methods, and selection of outcomes, and (c) how recognizing indigenous epistemological frameworks is necessary for understanding interventions and for guiding research to establish evidence” [17]. In the case of *My Journey*, the culture cube serves as the empirical framework from which we were able to articulate how culture informed the development of this program, how culture was infused into the development of our evaluation tools and methods, and how local Indigenous ways of knowing and being are integrated into the conceptual underpinnings of *My Journey* and serve as the community-defined evidence for determining the effectiveness of the program (see Section 3.1 below). 

### 1.3. Formative Research

Program development began with Northern Plains Native American community members, who were involved in all stages of the research process, including the selection of appropriate research design and methodology [28,29,30]. We also consulted community advisory boards (CABs) consisting of local residents of various ages, genders, and backgrounds about the creation of evaluation tools and surveys, establishing appropriate incentives as well as determining where, with, and by whom the program should be implemented. The formative research was also used to better understand sources of resilience, community attitudes and norms related to youth’s sexual behavior, and to garner input from the local community on what they would like to see included in a teen pregnancy prevention program.

This research included 24 focus groups with 185 urban and reservation-based Northern Plains Native youth (both parents and nonparents) and elders, as well as 20 interviews with local health care providers and school personnel. Questions focused on facilitators and barriers to teen pregnancy prevention, reasons youth have/do not have sex and use/do not use contraception, attitudes toward teen pregnancy, and recommendations for the content and implementation of a teen pregnancy prevention program. A detailed description of the recommendations for program content and activities is published elsewhere [31]. The top reccomendations are detailed below with a description of how these findings were incorporated into the *My Journey* program (see in Section 1.4.1) and utilized to provide PBE (see Section 4). 

### 1.4. Development of My Journey

Curriculum development and pilot testing took place over 3.5 years from start to professional printing. The aim of *My Journey* is to equip students with information, tools, and skills that enable them to take charge of their decision-making. Through *My Journey*, youth consider how culture and environment play important roles in a person’s fundamental beliefs and values. *My Journey* has youth identify these factors and explore the effects their decisions have on their personal lives, futures, and goals as well as the effects their decisions have on the people in their lives and places in which they live. By equipping students with skills and self-efficacy to make healthy decisions and increasing their positive ethnic identity, we aim to foster their attainment of positive life goals through reducing risk behaviors that lead to unintended teen pregnancy. For more details on the curriculum content, including lesson objectives and activities, see the Appendix B. 

#### 1.4.1. Utilization of Formative Data

One of the highly innovative parts of *My Journey* is the incorporation of elements of Indigenous cultures throughout the curriculum, which is important to PBE to base programs on local Indigenous worldviews and ways of knowing and being. Culture was an important element uncovered in the needs assessment, which underscores the importance of using a framework such as the culture cube assess if culture is accounted for appropriately. As summed up by this urban female youth (non-parent), “……I think if people were more educated about their culture and like traditions, and how things used to be, and what should be carried on today, maybe they would focus more on that. And like, and maybe that would become more time-consuming and more of like a value too than, rather than choosing to be sexually active” [31]. Therefore, traditional Native American teachings (e.g., the four quadrants of the medicine wheel), storytelling, and the application of traditional cultural values (e.g., respect, humility, compassion, bravery, generosity, wisdom) were applied to healthy decision-making and discussions of local coming of age ceremonies. 

In terms of program content directly related to teen pregnancy prevention, formative research participants recommended that *My Journey* cover healthy relationships and contraceptive knowledge access and use. Particular attention is paid to communication modeling, negotiation, and refusal skills. Vignettes are included to reflect the wisdom of tribal elders and the lived experiences of current and former Native teen parents, given the top suggestion to include testimonials and highlight the impact of teen pregnancy. In addition, former teen parents are invited to talk to students about their experiences as young parents and discuss the impacts of teen pregnancy for both parents. 

Critical to the acceptability of the *My Journey* curriculum with youth was integrating hands- on activities. Some of these activities included traditional games (e.g., Wolf and the Hen, Hoop Pass) drawn from a physical activity kit and training outlining various Indigenous games from tribes throughout the U.S. to promote age and culturally attuned moderate to vigorous physical activities [32]. 

Information in *My Journey* is presented in an engaging manner, with interactive activities, videos, and journal entries. Much care is taken during the first session to set a welcoming atmosphere and safe space for dialogue and questions. This includes a discussion on what situations students’ disclosure of sensitive information that mandatory reporting would occur. Each lesson has references to standards covered by state and national health and sex education standards, as well as Oceti Sakowin Essential Understandings (local Native American cultural information appropriate for grade-level). Additional relevant activities are provided in the appendices for facilitators to conduct with the students prior to the implementation to build rapport, or as supplemental activities throughout the curriculum to reinforce understanding of key concepts. 

#### 1.4.2. Core Concepts of My Journey

Five core concepts are the foundation of the *My Journey* curriculum: ∙Medicine wheel (balance of body, mind, emotion, and spirit)∙Chain Reaction Model (yesterday, today, tomorrow)∙Traditional Native American values (respect, perseverance, fortitude, humility, honor, love, sacrifice, truth, compassion, bravery, generosity, wisdom, and calm/quiet)∙Sexual health (taking personal and sexual responsibility)∙Goal setting.

These core concepts, as well as the curriculum itself, can be adapted to fit the local cultural context in which the program is being implemented. A symbol (i.e., hands molding a piece of clay) signifies areas that facilitators should review while preparing their lesson plan for the day to determine if and how the information and examples outlined in the lesson might be modified to better reflect the culture and environment in which the youth live.

The first is the core concept of the medicine wheel (see Figure 2), an ancient symbol used by nearly all Indigenous peoples in North and South America [33]. While there are numerous ways this symbol and its meanings have been and continue to be expressed, it is commonly represented as a wheel divided into four quadrants, with different peoples attributing different gifts from the Creator to each of the these dimensions. The curriculum emphasizes that there is not one right way to portray or infer meaning from the medicine wheel: the orientation, colors, and concepts depicted, along with their meanings and position, is just one of many interpretations. Facilitators encourage students to visit with their relatives and to do their own research, to find out how their tribe and families depict this symbol, its meanings, and applications. For the purposes of this program, this symbol is a nonlinear model of human development emphasizing balance while developing equally the physical, mental, emotional, and spiritual aspects of knowing and being. In other words, it is a holistic way of living a healthy life by nurturing and keeping all of the aspects of self in balance.

As articulated within the Relational Worldview framework [34,35,36,37], the physical self (body) are the parts and processes that comprise our physiological functioning, including genetics, chemistry, neurology, and biology. The mental self (mind) encompasses cognitive functions such as reasoning, memory, attention, and language. The emotional self (emotion) pertains to a person’s awareness, management, and expression of their mood and feelings towards yourself and others. The spiritual self (spirit) refers to our values, beliefs, meanings, and paths toward virtuous relations with all things (translated in Lakota as mitákuye oyás’iŋ, or “we are all related”). The four quadrants are interrelated and interdependent, and conditions in any one of these four areas influence all of the other areas. To use the model properly, “you must visualize yourself in the center of the wheel, connected to all points by the power of your own will”. [33] (p. 40).

The second core concept is the Chain Reaction Model (see Figure 3), which, along with the medicine wheel and other core concepts, lays the foundation for healthy decision-making. This model describes how our cumulative individual and collective histories (our yesterday, translated in Lakota as ehaŋni) inform our present circumstances (today or lehani), and present us with an opportunity to influence our trajectory (tomorrow or tokata). This model emphasizes the use of intentional reflection and agency in the decision-making process, stressing the importance of aligning our actions with our goals and our values, keeping all aspects of the self (body, mind, emotion, spirit) in balance. Critical to the reflection on tomorrow, is consideration of the positive and negative consequences that may result from choices made.

The third core concept of traditional Native American values (see Figure 4) is, in our curriculum, understood as the heart of a people’s culture. These values are reflected in and derived from our way of life and are used as a way of discerning how to virtuously conduct one’s self. Throughout *My Journey*, these values are used as a means of grounding our actions in traditional teachings and are woven into stories, discussions, traditional and modern games, and other important concepts like kinship, social support, finding trusted adults, ethnic identity, pride, coming of age ceremonies, and communication skill building.

The fourth core concept is sexual health. This concept highlights the importance of taking personal and sexual responsibility, meaning aligning sexual behaviors with your values and being accountable for those actions. Relatedly, a variety of nonconsensual sex scenarios are also addressed, and students are given the opportunity to think through and discuss situations that can arise that oscillate between ethically questionable to dangerous and illegal.

The fifth core concept is goal setting. Youth develop short- and long-term goals, with step-by-step plans for accomplishing them. They revisit these goals throughout as a means of mitigating risky sexual health behaviors or any other behavior that might prevent them from reaching their full potential. 

All of the core concepts relate to one another, making it easier for students to reflect upon and apply all of these teachings to various circumstances they may face throughout their development. For instance, goal setting relates directly to the other core concepts in that the decisions they make regarding their sexual health are informed by their past experiences and may directly affect their ability to accomplish these goals as well as their physical, mental, emotional, and spiritual well-being. 

### 1.5. Present Study

The focus of the present study is to describe a pilot study of the *My Journey* program, including an in-depth process evaluation and a preliminary outcomes evaluation, and analysis of the practice-based evidence using the culture cube framework.

## 2. Materials and Methods 

Prior to conducting this study, all study procedures were approved by Sanford Research Institutional Review Board (#03-13-068), Great Plains Institutional Review Board (AAIRB#13-R-15AA), and by the local tribe through tribal resolution (SWO-13-074) and local research review board (Permit#: SWO2016-005A2) for the reservation site. 

In 2015, the first year the program was implemented, the curriculum consisted of 30 lessons. Due to the revisions process during the following summer, the curriculum was shortened to 28 lessons when implemented in the spring and fall of 2016. Each lesson was designed to take 30 minutes to deliver. Since school schedules can vary, and class periods typically range from 30–90 minutes, lessons can (and often were) combined to accommodate the class schedule. At the schools the curriculum was implemented, classes were broken down into 50–60 minute periods, and were conducted in cultural studies and health classes, as preferred by the schools’ principals. Given that testing and surveys occasionally cut into class time, class length ranged from 25 to 60 minutes, with the average length being 52 minutes (*SD* = 7.4). Throughout the three semesters the program was evaluated, 95% of the curriculum was implemented. The lessons were planned as part of the curriculum with the classroom teacher and principal before the semester started. The classroom teacher was present (or facilitating) for a majority of the lessons.

The two facilitators during this time were from the community and enrolled tribal members. One facilitator was involved since the conception and throughout the development of *My Journey*, including the formative research that informed the curriculum, lesson development, pilot testing, and revisions. Though not a formally trained educator, she was experienced working with youth. The other facilitator was a health, drama, and music teacher and received a one-day training in *My Journey* plus in-class assistance throughout implementation by the other facilitator.

### 2.1. Participants

Pilot data were collected during three semesters *My Journey* was offered (spring 2015, fall 2015, and fall 2016) at two tribal schools within one tribal nation. Class size ranged from 14 to 21 students. Of the 53 students who participated in *My Journey*, 45 (84.9%) during one of these semesters took part in this study (female = 22; male = 23). After normality testing, one outlier was removed from analyses. All of the students identified as Native American alone (*n* = 35) or in combination with one or more other races (*n* = 9; missing = 1). Though most were members of the local tribe (*n* = 28), six other tribal affiliations were also identified, and three listed broad affiliations (i.e., Dakota, Sioux), and eight were missing. Students were in 6^th^ (*n* = 12), 7th (*n* = 24), and 8^th^ (*n* = 9) grades, and ages ranged from 11 to 14 (*M* = 13.2 years). Demonstrating that nearly all of the students were living in poverty or low-income households, approximately 96% (*n* = 43) of students indicated they were eligible for free or reduced lunch. Additionally, 35.6% (*n* = 16) had a parent with a bachelor’s degree or higher, 24.4% (*n* = 11) completed high school, and 2.2% (*n* = 1), had less than a high school education. Most of the students (*n* = 17, 37.8%), however, didn’t know the educational attainment of their parents. 

### 2.2. Process Evaluation

Fidelity and implementation were monitored using an observation instrument adapted from Project Venture’s session fidelity monitoring logs and observation forms [38]. This instrument documented curriculum activities planned for the day, adherence to the lesson plan, descriptions and explanations of changes made, assessments of the quality of facilitation, observations of student engagement, reflections on what went well, and suggestions for improvement. A total of 48 fidelity and implementation report forms were completed by the program facilitators and observers. Research team staff, the classroom teacher, and facilitators (when not implementing the program), served as the observers. The quality of instruction was measured via five questions:Delivery: Outlined objectives/purpose, was knowledgeable of material, and stated instructions/conclusions clearly and concisely.Investment in students: Built student rapport, trust, support, and fostered open communication.Demonstrating cultural responsiveness: Respected all cultures, included culturally relevant examples, and corrected assumptions and stereotypes.Time management: Stayed on task and was mindful of time constraints.Classroom management: Managed group dynamics to keep students on task, focused, and compliant with class values.

Students’ receptiveness to the curriculum was measured via 4 questions:Understanding of key concepts: Demonstrated knowledge of material, aka “Aha!” moments.Investment in material: Devoted time, effort, and attention to material.Level of engagement: Actively participated in activity and/or discussion.Conduct: Compliant with class values.

Responses were recorded on a 5 point Likert scale (1 = poor; 5 = excellent), and space was provided for facilitators and observers to explain their rationale for each rating. Fidelity and implementation monitoring was completed for 79% of the 82 lessons implemented. Of those, 45% were observed by a member of the research team or classroom teacher. 

### 2.3. Outcome Evaluation

Student outcomes were assessed via paper-and-pencil pre- and post-test surveys at the beginning and end of the semester in which students participated in the program. Surveys were coded and entered into IBM SPSS software, with a data entry check on over 10% of the data.

#### 2.3.1. Consent

To gain parental consent for youth to participate in the study surveys, parents were asked to sign a study consent form prior to having their teens complete a baseline study survey. A variety of methods were used to request parental consent (in person, school open house, mail, forms taken home in backpacks). Groups that received an 80% consent return rate from parents (indicating yes or no) receive a pizza party. The active parental consent rate was 74% (*n* = 50). Of those students whose parents provided consent, 92% assented to participate in the study (*n* = 46). 

#### 2.3.2. Outcome Measures

The survey tools were developed amongst the research team across a period of several months. Measures were adapted from established scales with youth populations which closely reflected the *My Journey* curriculum core concepts and goals.

Decision-making skills. Eight items adapted from the Mincemoyer and Perkins [39] Making Decisions in Everyday Life Scale were used to evaluate decision-making skills. Questions explored various factors related to their general decision-making process (e.g., “Consider the risks and consequences of a choice before making a decision,” “Discuss choices with my parents or other adults”). Responses were recorded on a 4-point scale ranging from never (coded as 1) to always (coded as 4). 

Ethnic identity. To measure the subjective sense of membership to an ethnic group, we used an adaptation of Phinney [40] Multigroup Ethnic Identity Measure (MEIM), which has been widely used in psychological and behavioral health studies. This scale is preceded by a question to identify students’ ethnic group. Twelve close-ended items then assess *exploration* of (five items) and *affirmation, belonging, and commitment* to (seven items) their ethnic group on a 5-point scale ranging from 1 = NO! to 4 = YES!. Example items include, “I have spent time trying to find out more about being Native, such as its history, traditions, and language” and “I have a strong sense of belonging to my Native community”.

Prosocial connectedness. Prosocial connectedness was assessed using eight-items adapted from the parent-family and school-connectedness scales developed by Resnick et al. [41]. The 4-point response scale ranged from NO! (coded as 1) to YES! (coded as 4). Sample items include, “My family understands me,” “I feel close to my father or the person most like my father,” “Adults at school expect me to do well”.

Reasons for having or not having sex. We explored reasons for having or not having sex using measures developed by Coyle et al. [42]. *Reasons for not having sex* consisted of six items that examined various motivations for sex refusal (e.g., “I would not have sex because I do not want to have a baby right now,” “My family would be disappointed if I had sex at my age”). *Reasons for having sex* consisted of seven items (e.g., “So that my boyfriend or girlfriend would not break up with me,” “To feel more loved and accepted”). Response options ranged from 1 = NO! to 4 = YES!. 

Sex refusal self-efficacy. We used an adaptation of the self-efficacy to refuse sexual activity measure developed by Coyle, Kirby, Marín and Gómez [42] for the Draw the Line/Respect the Line sexual risk behavior prevention intervention. This scale consists of six items that asked participants to imagine they were with someone they really liked and to rate their personal agency in sexual negotiation (e.g., “Could you stop yourself form having sex if the person said they would break up with you unless you had sex with them?” “Could you stop them if they wanted to touch your private parts below the waist?”). The 4-point response scale ranged from NO! (coded as 1) to YES! (coded as 4). 

#### 2.3.3. Planned Analyses

Data management and analysis were conducted using IBM SPSS software. Various descriptive statistics were employed to clean data and describe sample characteristics (e.g., frequencies, means, standard deviations, including using chi-square tests for categorical variables and t-tests for continuous variables). Paired t-tests were used to assess change from pre-test to post-test on study outcome measures. 

### 2.4. Practice-based Evidence Evaluation

In order to assess the practice-based evidence for *My Journey*, we turned to a recently developed conceptual tool called the culture cube, an evaluation measure developed specifically for PBE prevention programs. Aligning with one of the cube’s articulated functions, use of the cube allowed us to clearly identify the cultural framework of *My Journey* by highlighting “the links between cultural beliefs and values, community needs, and intervention design” [17]. This evaluation was completed post-pilot study, as publication of the cube was released in the summer of 2018.

## 3. Results

### 3.1. Process Evaluation

The process evaluation examining feasibility and acceptability of *My Journey* focused on fidelity to the curriculum and observational suggestions for curriculum revisions and facilitation improvements. As seen in Table 1, facilitators scored highest in cultural responsiveness (*M* = 4.33, *SD* = 0.63, range = 2–5) and lowest in time management (*M* = 3.94, *SD* = 0.85, range = 3–5). Cultural responsiveness (i.e., demonstrating respect for all cultures, using culturally relevant examples, and correcting assumptions and stereotypes) ranked high. Time management may have been lower due to it being the first time activities were piloted in a classroom setting, and some activities took longer or shorter than anticipated.

Students scored highest in investment in material (*M* = 3.90, *SD* = 0.68, range = 2–5) and lowest in conduct (*M* = 3.73, *SD* = 0.74, range = 2–5). Considerable time, effort, and attention was devoted to the development of culturally aligned, hands-on activities and modifying content based off of the feedback provided by facilitators and observers, which was likely why higher levels of investment and engagement were observed. In spite of this, facilitators noted some students (or even classes) being especially hyperactive, which is not uncommon, and consequently having to consistently address disruptive behaviors and revisit class values and expectations to address these issues.

In their reflections, facilitators and observers often noted, in addition to recommendations for program changes, student progress in a number of areas. Specifically, facilitators detected gains in students’ sexual health knowledge and comfort discussing these topics. For example, one observer noted, “The students engaged in class discussion. They were less embarrassed when talking about sexual responsibility”. They also noted how students made effective applications of the core concepts to their decision-making process (e.g., “The curriculum included many scenarios to discuss. This helped the students to become prepared for the different situations they may be faced with”). Culture was also noted as a strength of the curriculum. For example, observers noted, “Students seem to be impressed with the cultural aspects of *My Journey*. I feel it gives them a sense of validation” and “The kids are culturally mindful and are able to have classroom discussions”.

Many of the observer’s comments focused on the students’ acceptability of the curriculum, namely the engagement and enjoyment of the lesson activities. For example, observers noted, “The kids really liked the Hoop Pass game. They seem like they are beginning to understand the Chain Reaction Model”. Aspects of the activities students appeared to particularly enjoy were the traditional games and STI transmission activity, as seen in the following example quotes:∙“I think the kids learned a lot about STIs and their transmission. The students were very engaged with the STI activity. The students were able to sit and learn about the different types of STIs.”∙“The students really liked the discussion on this [birth control trivia]. . . . They also like talking about personal experiences with people they know or what they have read in the media. . . . The students said they learned a lot about pregnancy today.”

Observations of *My Journey* also noted many strengths of the program, particularly in terms of its acceptability. For example, facilitators observed high levels of student investment and engagement in the course material, attributing this, in part, to the cultural aspects of the program and kinesthetic teaching methods, such as traditional games. Additionally, process evaluation findings provide evidence that *My Journey* effectively worked towards achieving the outcomes of increasing ethnic identity and pride and sexual health knowledge and responsibility. 

Of the three semesters *My Journey* was implemented, program fidelity was highest in the second semester (spring of 2016). This may be due in part to this being the second semester the facilitator was teaching this class, with greater experience, familiarity, and comfort with the curriculum. Fidelity was lowest the third semester (fall of 2016). This may reflect the fact that this was the first semester *My Journey* was taught at this particular school, and this facilitator’s first semester with the curriculum. However, had evaluation of the program continued, we would have likely observed fidelity increasing over time.

### 3.2. Outcome Evaluation

Table 2 depicts the social and behavioral outcomes for the youth participants’ pre and post intervention. As shown by the t-test results, at post-intervention, participants demonstrated higher levels of self-efficacy t = −3.567, (*M* = 3.69) compared to pre-intervention (*M* =3.47). Although not significant, the decision-making skills and reasons for *not* having sex also increased while reasons for *having* sex decreased. Somewhat unexpectedly, we did not find a significant increase in ethnic identity or its subscales of affirmation/belonging and exploration. Similarly, there was no change found on prosocial connectedness.

### 3.3. Practice-based Evidence Evaluation

By aligning the elements of *My Journey* with the 6 aspects of the culture cube (project, persons, place, culture, causes, & changes), we were able to determine that *My Journey* is a community-defined, effective, culturally integrated prevention curriculum for Native youth. Table 3 details the cultural elements of *My Journey* as aligned with the visible and invisible elements of the cube, which was helpful for reflecting on the effectiveness of the cultural elements of the program, such as the Chain Reaction Model, the application of the medicine wheel, and the various activities that draw upon Indigenous cultures and values in order to encourage positive sexual health decision-making. 

## 4. Discussion

We created *My Journey*, a culturally attuned sexual health curriculum for Native youth, to address the need for teen pregnancy prevention programs for this population. The results of our pilot study indicate that, not only are the facilitators pleased with the curriculum and its application within tribal youth classrooms (grades 6–8), but *My Journey* has a positive effect on the self-efficacy of Native teens. Decision-making skills and a respect for one’s self and others are encouraged through the emphasis *My Journey* places on Indigenous cultures and values, a major component of the curriculum. The lack of significant changes in the study’s quantitative measure of ethnic identity may be due to a ceiling effect, as the students reported high levels of this measure of ethnic identity at the pre-test assessment. Support for this interpretation has been found in a previous study which a youth program failed to create a strong Native American identity, likely because it was attracting Native youth with an already high ethnic identity [45]. The present study may have similar underpinnings, as these tribal school youth have a relatively high exposure to cultural elements throughout school and their community. Although the documented quotes from the process evaluation do not speak directly to ethnic identity being built, we received a great deal of informal feedback from facilitators that traditional values were a unique and impactful component of *My Journey*.

Developed through CBPR methods and analyzed with a PBE cultural lens, *My Journey* draws on the rich history and depth of Indigenous cultures to engage and inform Native youth, and as such, culture is one of the strongest, most innovative aspects of the *My Journey* curriculum. After the application of the culture cube, our confidence in the cultural underpinnings of *My Journey*, and the community-defined evidence for the curriculum has been bolstered. 

Increasingly, researchers are beginning to acknowledge the need for innovative program design, implementation, and evaluation when working with culturally and ethnically diverse populations. Many have called for a reevaluation of what constitutes “evidence” when serving non-majority populations, since “evidence” often indicates the use of randomized controlled trials, which require higher numbers of participants than certain communities can provide, as well as a “controlled” environment, which negates the rich, cultural and extemporaneous nuances of the human experience. Because *My Journey* was developed for a specific population and piloted with a small number of students within that group, it was expected that the numbers might not immediately align with Western definitions of significance. But, as a growing number of researchers have argued, in order to develop any culturally attuned, innovative program for minority populations, we must seriously consider strategies and evidence that may or may not align with Western scientific methods [18,46]. Some have even gone so far as to suggest that the guidelines for evaluating innovative programs like *My Journey* should be changed [18] to utilization of the culture cube or similar methodologies. 

## 5. Conclusions

We are encouraged by positive facilitator insights and student growth due to *My Journey*. Our hope is to expand, sustain, and adapt *My Journey* by securing funding that will allow us to implement the program in other communities and schools, including elementary schools, as requested by the community. Expanding the program beyond Northern Plains tribes would require considerable time, effort, and resources, which may not be readily available. This is especially true for many rural, reservation schools with limited resources whose students and communities continue to face the long-term effects of historical trauma. In future iterations of the program, community input will continue to be critical in order to adapt and create culturally aligned evaluation methods and tools and assess the community’s awareness and satisfaction with the program, how it is being implemented, and its’ impact. We encourage researchers to expand upon these efforts by developing more diverse research teams, including local educators, students, elders, curriculum writers, and researchers in planning and implementation. This will be particularly important when *My Journey* is adapted and implemented in communities outside of the ones in which it was developed, as this would require adjustments to the PBE used to establish the efficacy of the program. Each tribal community and school is unique and requires thoughtful attention and respect in order to develop culturally attuned PBE interventions. The utilization of the culture cube is one way researchers can develop and/or evaluate PBEs, hopefully leading to a future in which the creation of innovative, culturally attuned PBEs for diverse, non-majority communities is accepted in the broader scientific community.

## Figures and Tables

**Figure 1 ijerph-16-00470-f001:**
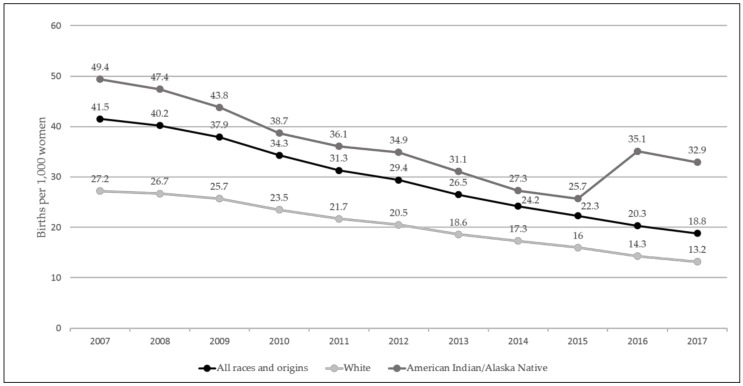
Teen (15–19 year olds) birth rate by race of mother (2007–2017) [1,5].

**Figure 2 ijerph-16-00470-f002:**
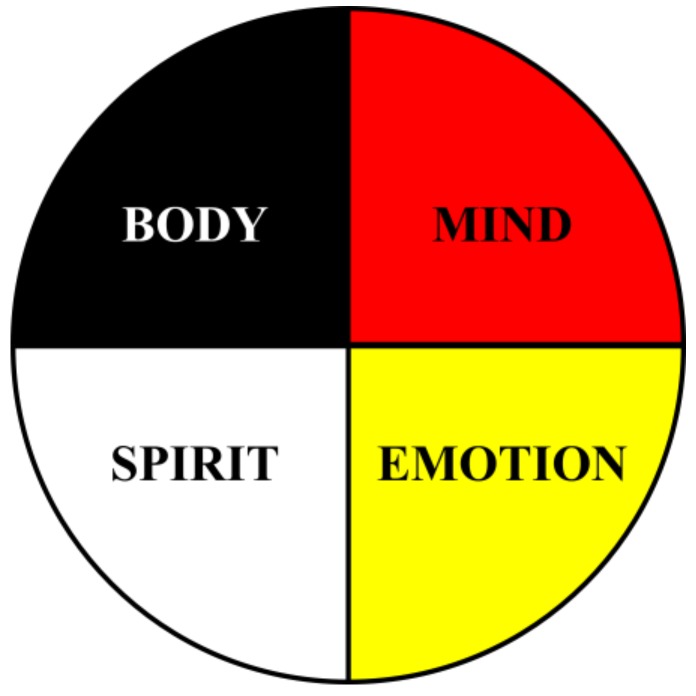
Medicine Wheel.

**Figure 3 ijerph-16-00470-f003:**
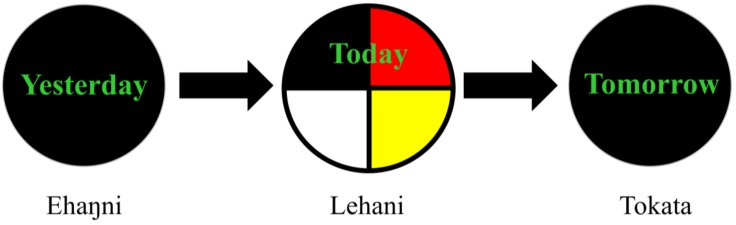
Chain Reaction Model (with translations of yesterday, today, and tomorrow in Lakota).

**Figure 4 ijerph-16-00470-f004:**
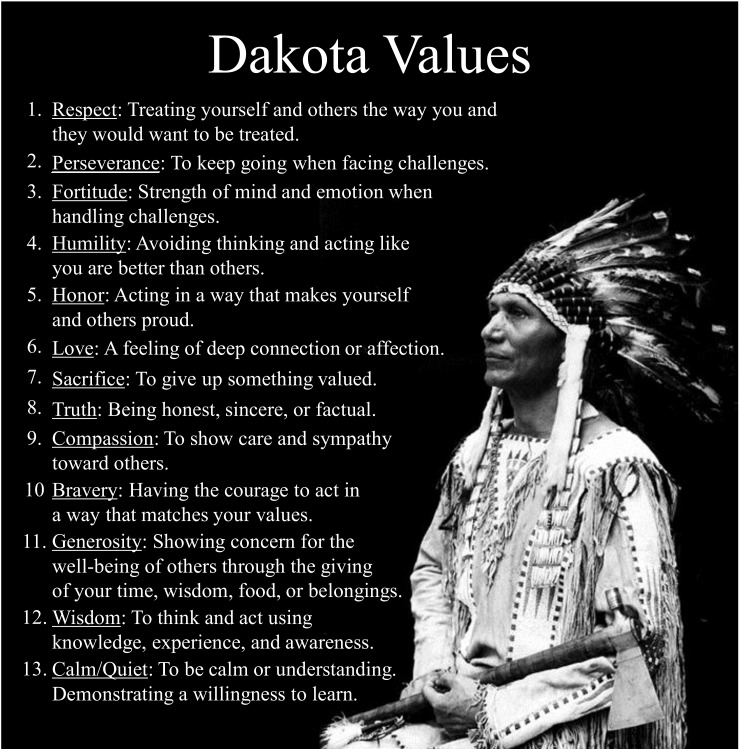
Traditional Dakota Values Poster.

**Table 1 ijerph-16-00470-t001:** Facilitation and Student Observations (*n* = 48 completed observation forms).

Topics	Mean	SD	Range
*Facilitation Observations*			
· Delivery	4.11	0.82	2.40–5
· Investment in students	4.24	0.74	2.00–5
· Cultural responsiveness	4.33	0.63	2.00–5
· Time management	3.94	0.85	3.00–5
· Classroom management	4.14	0.71	2.50–5
*Student Observations*			
· Understanding of concepts	3.84	0.66	2.25–5
· Investment in material	3.90	0.68	2.00–5
· Level of engagement	3.86	0.76	2.00–5
· Conduct	3.73	0.74	2.00–5

**Table 2 ijerph-16-00470-t002:** Comparison of Pre-Test and Post-Test Scores on Scales of Interest (*n* = 45).

Scale	Possible Range	Cronbach α Pre/Post	Pretest Mean (SD)	Posttest Mean (SD)	Change Mean (SE – Pre/Post)	Paired t-Test
Decision-making skills	1–4	0.82/0.87	2.54 (0.56)	2.63 (0.63)	0.09 (0.09/0.10)	−0.94
Ethnic identity	1–4	0.86/0.94	3.38 (0.41)	3.34 (0.55)	−0.04 (0.06/0.08)	−0.15
Affirmation belonging subscale	1–4	0.89/0.94	3.49 (0.47)	3.45 (0.56)	−0.04 (0.07/0.09)	−0.13
Exploration subscale	1–4	0.70/0.84	3.22 (0.50)	3.20 (0.60)	−0.02 (0.08/0.09)	−0.17
Prosocial connectedness	1–4	0.85/0.78	3.06 (0.53)	3.06 (0.48)	0 (0.08/0.07)	−0.07
Reasons for not having sex	1–4	0.84/0.80	3.54 (0.59)	3.59 (0.51)	0.05 (0.09/0.08)	−0.67
Reasons for having sex	1–4	0.88/0.90	1.39 (0.49)	1.34 (0.53)	−0.05 (0.08/0.08)	1.35
Sex refusal self-efficacy	1–4	0.79/0.90	3.47 (0.49)	3.69 (0.50)	0.22 (0.08/0.08)	−3.57 *

* *p* < 0.05.

**Table 3 ijerph-16-00470-t003:** Application of the Culture Cube Framework for My Journey.

The Observable	
*Project*: What is the activity or the community-defined practice/intervention?	Community suggestions for a teen pregnancy prevention program, include (in order of the frequency each was mentioned) [31]: 1) *Testimonials/guest speakers* (highlighting teen pregnancy impact): The program includes an in-person interview with a current or former teen parent in which youth can also ask the speaker questions about their experiences and/or seek advice. 2) *Cultural education*: Translations of traditional values, ceremonies, and other cultural teachings are provided in the local tribal language along with Native American history, cultural/ethnic identity and pride, ceremonies, and traditional values discussions and activities. 3) *Hands-on activities*: Experiential and kinesthetic learning techniques are used throughout. 4) *Sex/reproductive health information*: Comprehensive sexual and reproductive health information provide youth with the knowledge and resources to make healthy, informed sexual decisions. 5) *Strengths-based approach*: The curriculum reflects a positive youth development framework by promoting empowerment in sexual health decision-making and engaging youth in prosocial behaviors [31].
*Persons*: Who will be involved in delivering and participating in *My Journey*, and what are they doing?	1) Suggestions for program delivery stressed the importance of the program being facilitated by dedicated, respected, and trustworthy local tribal members that youth could relate to. Both of our facilitators fit these criteria. 2) In the focus groups and interviews, community members suggested the program be designed for elementary and middle school aged youth and, as a result, was implemented with 6^th^, 7^th^, & 8^th^ graders.
*Place*: Where does *My Journey* take place in terms of the organizational and/or community setting and geographic location and why is this important?	1) Suggestions for program delivery stressed the desire for a school-based program for elementary and middle school aged youth. Therefore, the program was implemented in two local, tribally run schools. One is a preschool through 8^th^ grade school and the second is a K-12 school. 2) A local depiction and interpretation of the medicine wheel was used as one of the core concepts as well as local traditional values. For example, Dakota values included the virtue of calm/quiet, which was not included in Lakota values, as suggested by a local elder who reviewed the core concepts before implementing in the local school district.
**The Invisible**	
*Culture*: How does *My Journey* reflect the cultural values, practices, and beliefs of NA communities?	1) Use of the medicine wheel highlights the four dimensions of wellness recognized historically by Native American communities. 2) Games and activities emphasize traditional values and expectations that can be applied to sexual health decision-making. 3) The Chain Reaction Model—the concepts of yesterday, today, and tomorrow—brings into focus the traditional values of NA culture (yesterday), how those values can be used to make healthy decisions (today), and how those decisions can help actualize their goals for themselves and future generations (tomorrow). 4) Interwoven throughout are various applications of Indigenous cultures: historic and modern contributions to society, coming of age ceremonies, kinship, interconnectedness, etc.
*Causes*: What are the problems the project is trying to address? How did it start and why? How are causes understood in (a) a historical context, (b) through the lens of the community’s values, and (c) things that concern or bother the community?	Native Americans experience higher teen pregnancy rates compared to other racial/ethnic groups. Community perceptions on the context and causes of this are varied [11,43]. 1) The belief that children are sacred gifts from the Creator, that pregnancy at any age is a blessing and therefore desirable. 2) Native girls are encouraged by their parents to have children at a young age, with the understanding that their parents will take care of their grandchild(ren), as their parents did. 3) The bulk of parenting responsibilities fall on teen mothers, which the community recognized as a problem. Teen fathers did not feel responsible for using contraception or taking care of or have a relationship with their children, given that many of them grew up in households without a dad. 4) Native boys learn that having sex with multiple partners is equated with being a man. They are taught to be proud of getting someone pregnant, as it is a sign of their virility and a way of keeping the tribal population thriving, which is also indicative of a relationship between high teen pregnancy rates and historical trauma. 5) Native girls felt pressured and even obligated to have sex to please their man and were sometimes taken advantage of when under the influence of drugs or alcohol, indicating that some teen pregnancies are the result of forced sex.
*Changes*: From our cultural perspective, what are the desired outcomes of *My Journey* for our community? We will see more of … and less of …	*My Journey* encourages healthy decision-making, and will ideally lead to the following outcomes, above and beyond a reduction in teen pregnancy, outlined by the community [11,31,43,44]: 1) Increased ethnic identity and pride 2) Increased prosocial connectedness 3) Increased sexual health knowledge and responsibility 4) Increased sex refusal self-efficacy

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
