# Peer review of "My Journey: Development and Practice-Based Evidence of a Culturally Attuned Teen Pregnancy Prevention Program for Native Youth"

_ijerph, 2019, doi:10.3390/ijerph16030470_

Round 1

Reviewer 1 Report

In general, this is a very important article, presenting a participatory intervention which incorporate traditional values and concepts of the target population.

The following comments should be taken into account for improving the messages:

It could have been a good example of an evidence based practice, to be used by others, however, some details should become clearer.

Who were the facilitators of the sessions – this should be specified in the methodology and not only in the results. It appears there (r. 370) that 2 facilitators were acting in the program. Their qualifications are not mentioned or whether they received a special training for that and from whom.  

Were these lessons planned in the curriculum? Was the classroom teacher present during these sessions?

“Fidelity and implementation were monitored using an observation instrument adapted from Project Venture’s session fidelity monitoring logs and observation forms [38]” – as this reference is only a personal communication, the authors should have given more details. The main point which is missing is who filled in the monitoring logs (there is information about the observation part).

Table 1 presents facilitation and student observations which were measured via nine questions evaluating the quality of facilitators’ instruction (five questions) and students’ receptiveness to the curriculum. The “(N=48)” which appears in the title does not have any reflection in the text – is that the number of observations? It is not clear at all.

There should be a better differentiation between methods, results and discussion. There is some discussion with the results and there are methods in the discussion. Table 3 mainly summarizes the principles of the intervention (which are mentioned in the methodology).

Author Response

Thank you for seeing the potential contribution of this manuscript. Your feedback was very helpful to improving the quality of the paper. Please find our responses to your comments below in bold:

Who were the facilitators of the sessions – this should be specified in the methodology and not only in the results. It appears there (r. 370) that 2 facilitators were acting in the program. Their qualifications are not mentioned or whether they received a special training for that and from whom.  

-We have moved information on the facilitators to the methodology and added additional information on their qualifications and training (p. 8)

Were these lessons planned in the curriculum? Was the classroom teacher present during these sessions?

-Text was added (p. 8) to detail that the lessons were planned as part of the curriculum with the classroom teacher and principal before the semester started. The classroom teacher was present (or facilitating) for a majority of the lessons.

“Fidelity and implementation were monitored using an observation instrument adapted from Project Venture’s session fidelity monitoring logs and observation forms [38]” – as this reference is only a personal communication, the authors should have given more details. The main point which is missing is who filled in the monitoring logs (there is information about the observation part).

-Additional information on the fidelity monitoring log and observers was included in Section 2.2 (pp. 8-9). We also took out the overall means/SD/range scores because we do not believe the original authors did not see these separate aspects of facilitation and student observations as a scale.

Table 1 presents facilitation and student observations which were measured via nine questions evaluating the quality of facilitators’ instruction (five questions) and students’ receptiveness to the curriculum. The “(N=48)” which appears in the title does not have any reflection in the text – is that the number of observations? It is not clear at all.

-N=48 is the number of completed observation forms, which has been clarified in the text and table.

There should be a better differentiation between methods, results and discussion. There is some discussion with the results and there are methods in the discussion. Table 3 mainly summarizes the principles of the intervention (which are mentioned in the methodology).

-We have reworked the some of the content of the methods, results, and discussion. The changes made mostly encompassed clarifying the use of the culture cube analysis as a Practice-based Evidence Evaluation, adding text in the methods and results (moving Table 3 up to the results).

Reviewer 2 Report

In the present manuscript, Kenyon et al. have summarized a culturally attuned program that decreases teen pregnancy rates in Native American communities. This study is timely because recent data indicate that teen birth rate for Native Americans is nearly twice as high as national averages. To reduce this disparity, the authors have developed a new curriculum, My Journey, to prevent teen pregnancy in such communities. The results are encouraging in that My Journey has a significant effect on the self-efficacy of Native American teens, and teachers prefer this new curriculum. These exciting results lend support to the broader implementation of My Journey.

Section 3.2. Outcome evaluation

Could the authors expand the present description of results to include more necessary information for understanding? For instance, why the authors focused on the eight measures in Table 2? And more description of the statistics. 

Author Response

Could the authors expand the present description of results to include more necessary information for understanding? For instance, why the authors focused on the eight measures in Table 2? And more description of the statistics.

-The entire survey included several other measures (intrapersonal skills, social skills, emotional regulation, depression, empowerment, hope, perceived utility value of school, peer victimization, peer normative beliefs favoring sex, exposure to situations that could lead to sexual behavior, sexual behaviors, contraceptive use, associated risky behaviors (i.e., alcohol, cigarette, and marijuana use), and program satisfaction. We have included more information on how the survey was developed and range of measures that were included (p. 9). In some cases, the internal reliability (alphas) were extremely low, introducing doubt about the accuracy of the tool for our population (e.g., empowerment scale). In other cases, the scales were unusable because responses were so skewed, or low (e.g., sexual behaviors, contraception use, risk behaviors). The eight measures included in the present study manuscript represent those measures most closely related with the curriculum’s core goals. The pre/post t-test results for the other usable measures not included in the table were non-significant. More information on the statistics was included to detail the non-significant findings.

Reviewer 3 Report

The authors present original data from the development and pilot testing of My Journey, a culturally attuned teen pregnancy prevention program for early adolescent Native American youth. Given the disparities faced by Native American youth in terms of sexual and reproductive health, and the dearth of curricula explicitly developed or adapted for Native youth, the study contributes to our understanding of how to effectively develop and evaluate a culturally attuned sexual health education intervention for this underserved, vulnerable population. The authors delineate the importance of developing practice-based evidence utilizing CBPR principles to develop and evaluate culturally acceptable interventions, and provide a detailed application of the culture cube framework to indigenous adolescent sexual and reproductive health, which provides a useful template for other investigators/practitioners. Overall, the study appears to have been thoughtfully designed and well-implemented. Outcome measures for the pilot study are age-appropriate and the sample size is adequate for an acceptability/feasibility trial. Overall, the manuscript is well-written with detailed examples to illustrate conceptual constructs. The following minor comments are offered to strengthen the manuscript.

1.4.1. Utilization of Formative Data, p.5, line 191. It would be helpful to provide a little more description of the take-home assignments. Did these involve conversations with parents or other adults? If not, how was parent-child communication integrated into the curriculum, if at all, since this is a common element of early adolescent sexual health education curricula?

2. Materials and Methods, p.8, lines 288-289. In what instruction areas (e.g., social studies, health education) was My Journey taught? 28-30 lessons is a lot of additional material to incorporate into one semester. Information regarding “best fit” would be helpful to other investigators/practitioners in adolescent sexual health.

2.1. Participants, p.8, lines 298-330. The sentence regarding levels of parental education is awkward to read, fix missing parentheses and grammar.

3.1. Results. Process evaluation, p.10, line 375. Student observations – M and SD provided in parentheses in text do not align with data provided in Table 1, is this a typo? Or clarify how calculated. Also, provide information on the completion rate for take-home assignments and any data on instructor/youth reactions to take-home assignments.

3.1. Results. Process evaluation, p.11, lines 406-408. Clarify how “process evaluation findings provide evidence that My Journey effectively worked towards achieving the outcomes of increasing ethnic identity and pride…” This is not self-evident, apart from a couple of example quotes.

3.2. Outcome evaluation, p. 11. Include a sentence summarizing impact on ethnic identity and subscales.

4. Discussion, p. 11. Include some discussion on the apparent limited/null effect on ethnic identity in the outcome results, maybe a ceiling effect? Reiterate process evaluation data regarding positive influence of ethnic identity.

Appendix A. Consider including placement and content of take-home assignments in the My Journey table of contents.

Author Response

Thank you for seeing the potential contribution of this manuscript, and were excited to hear you found the study “contributes to our understanding of how to effectively develop and evaluate a culturally attuned sexual health education intervention for this underserved, vulnerable population”. Your feedback was very helpful to improving the quality of the paper. Please find our responses to your comments below in bold:

1.4.1. Utilization of Formative Data, p.5, line 191. It would be helpful to provide a little more description of the take-home assignments. Did these involve conversations with parents or other adults? If not, how was parent-child communication integrated into the curriculum, if at all, since this is a common element of early adolescent sexual health education curricula?

-The take-home assignments evolved over the My Journey development period. During the early stages of the pilot study’s implementation period there were take home assignments (Ex1: talk with a trusted adult about the Native American values discussed in class, which are practiced daily, which do you feel you need to work on. Ex2: ask parent/friend/trusted adult about a specific time that made him/her proud of being Native, how did this impact their decisions made at the time, how does that part of this person’s background still impact him/her today?). However, due to difficulties in return of homework similar to other youth programs (e.g., Grossman et al., 2013 Journal of School Health) these assignments were modified to become in-class journaling assignments. Since that is the direction the current curriculum has taken, we have removed the reference to take-home assignments in the text.

2. Materials and Methods, p.8, lines 288-289. In what instruction areas (e.g., social studies, health education) was My Journey taught? 28-30 lessons is a lot of additional material to incorporate into one semester. Information regarding “best fit” would be helpful to other investigators/practitioners in adolescent sexual health.

- We have added details on the instruction areas in which My Journey was taught (p. 8).

2.1. Participants, p.8, lines 298-330. The sentence regarding levels of parental education is awkward to read, fix missing parentheses and grammar.

- We have edited this sentence.

3.1. Results. Process evaluation, p.10, line 375. Student observations – M and SD provided in parentheses in text do not align with data provided in Table 1, is this a typo? Or clarify how calculated. Also, provide information on the completion rate for take-home assignments and any data on instructor/youth reactions to take-home assignments.

- We have updated the text and table. We did not collect information on the completion rate for take-home assignments. As noted above, facilitators reported challenges getting students to return their take-home assignments. Based on this feedback, the curriculum was revised to move from take-home assignments to in-class journaling. Therefore, this reference to take-home assignments was removed from the text. 

3.1. Results. Process evaluation, p.11, lines 406-408. Clarify how “process evaluation findings provide evidence that My Journey effectively worked towards achieving the outcomes of increasing ethnic identity and pride…” This is not self-evident, apart from a couple of example quotes.

- We have included additional text clarifying the cultural aspects of the curriculum, careful not to overstate that ethnic identity was built through the curriculum (p. 11).

3.2. Outcome evaluation, p. 11. Include a sentence summarizing impact on ethnic identity and subscales.

- We have included additional text in the outcomes evaluation section, summarizing intervention impact on ethnic identity (p. 11).

4. Discussion, p. 11. Include some discussion on the apparent limited/null effect on ethnic identity in the outcome results, maybe a ceiling effect? Reiterate process evaluation data regarding positive influence of ethnic identity.

- We have included additional discussion on ethnic identity and the possible ceiling effect, citing a previous study with Native American youth (p. 14).

Appendix A. Consider including placement and content of take-home assignments in the My Journey table of contents.

- See previous response to comment 3.1. We have added section 1.4.3 to detail the take-home assignments, including examples. Since the pilot study, facilitators have moved towards incorporating the take home assignments into the class period, therefore, we prefer not to incorporate them into the table of contents in the appendix.